# RegQ: Convergent Q-Learning with Linear Function Approximation using Regularization

## Abstract

Q-learning is widely used algorithm in reinforcement learning community. Under the lookup table setting, its convergence is well established. However, its behavior is known to be unstable with the linear function approximation case. This paper develops a new Q-learning algorithm, called RegQ, that converges when linear function approximation is used. We prove that simply adding an appropriate regularization term ensures convergence of the algorithm. Its stability is established using a recent analysis tool based on switching system models. Moreover, we experimentally show that RegQ converges in environments where Q-learning with linear function approximation has known to diverge. An error bound on the solution where the algorithm converges is also given.

## 1 Introduction

Recently, reinforcement learning has shown great success in various fields. For instance, Mnih et al. (2015) achieved human level performance in several video games in the Atari benchmark (Bellemare et al., 2013). Since then, researches on deep reinforcement learning algorithms have shown significant progresses (Lan et al., 2020; Chen et al., 2021). For example, Badia et al., 2020 performs better than standard human performance in all 57 Atari games. Schrittwieser et al., 2020 solves Go, chess, Shogi, and Atari without prior knowledge about the rules. Although great success has been achieved in practice, there is still gap between theory and the practical success. Especially when off-policy, function approximation, and bootstrapping are used together, the algorithm may diverge or show unstable behaviors. This phenomenon is called the deadly triad (Sutton and Barto, 2018). Famous counter-examples are given in Baird, 1995; Tsitsiklis and Van Roy, 1997.

For policy evaluation, especially for temporal-difference (TD) learning algorithm, there has been several algorithms to resolve the deadly triad issue. Bradtke and Barto, 1996 uses the least-square method to compute a solution of TD-learning, but it suffers from $O(h^2)$ time complexity, where $h$ is number of features. Maei, 2011; Sutton et al., 2009 developed gradient descent based methods which minimize the mean square projected Bellman error. Ghiassian et al., 2020 added regularization term to TD Correction (TDC) algorithm, which uses a single time scale step-size. Lee et al., 2021 introduced several variants of the gradient TD (GTD) algorithm under control theoretic frameworks. Sutton et al., 2016 re-weights some states to match the on-policy distribution to stabilize the off-policy TD-learning. Diddigi et al., 2019 uses $l_2$ regularization to propose a new convergent off-policy TD-learning algorithm. Mahadevan et al., 2014 studied regularization on the off-policy TD-learning through the lens of primal dual method.

First presented by Watkins and Dayan, 1992, Q-learning also suffers from divergence issues under the deadly triad. While there are convergence results under the look-up table setting (Watkins and Dayan, 1992; Jaakkola et al., 1994; Borkar and Meyn, 2000; Lee and He, 2019), even with the simple linear function approximation, the convergence is only guaranteed under strong assumptions (Melo et al., 2008; Lee and He, 2019; Yang and Wang, 2019).

The main goal of this paper is to propose a practical Q-learning algorithm, called regularized Q-learning (RegQ), that guarantees convergence under linear function approximation. We prove its convergence using the ordinary differential equation (O.D.E) analysis framework in (Borkar and Meyn, 2000) together with the switching system approach developed in Lee and He, 2019. As in Lee and He, 2019, we construct upper and lower comparison systems, and prove its global asymptotic stability based on switching system theories. Compared to the standard Q-learning in (Watkins and Dayan, 1992), a difference lies in the additional $l_2$ regularization term, which makes the algorithm relevantly simple. Moreover, compared to the previous works in Carvalho et al. (2020); Maei et al. (2010), our algorithm is single time-scale, and hence, shows faster convergence rates experimentally. Our algorithm directly uses bootstrapping rather than circumventing the issue in the deadly triad. Therefore, it could give a new insight into training reinforcement learning algorithms with function approximation without using the so-called target network technique introduced in Mnih et al., 2015. The main contributions of this paper are summarized as follows:

1. A new single time-scale Q-learning algorithm with linear function approximation is proposed.

2. We prove the convergence of the proposed algorithm based on the O.D.E approach together with the switching system model in Lee and He, 2019.

3. We experimentally show that our algorithm performs faster than other two time-scale Q-learning algorithms in Carvalho et al. (2020); Maei et al. (2010).

**Related works**:

Several works (Melo et al., 2008; Lee and He, 2019; Yang and Wang, 2019) have relied on strong assumptions to guarantee convergence of Q-learning under linear function approximation. Melo et al., 2008 adopts an assumption on relation between behavior policy and target policy to guarantee convergence, which is not practical in general. Lee and He, 2019 assumes a strong assumption to ensure the convergence with the so-called switching system approach. Yang and Wang, 2019 has a stringent assumption on anchor state-action pairs.

Motivated by the empirical success of the deep Q-learning in Mnih et al., 2015, recent works in Zhang et al., 2021; Carvalho et al., 2020; Agarwal et al., 2021; Chen et al., 2022 use the target network to circumvent the bootstrapping issue and guarantees convergence. Carvalho et al., 2020 uses a two time-scale learning method, and has a strong assumption on the boundedness of the feature matrix. Zhang et al., 2021 uses $l_2$ regularization with the target network, while a projection step is involved, which makes it difficult to implement practically. Moreover, it also uses a two time-scale learning method. Chen et al. (2022) used target network and truncation method to address the divergence issue. Agarwal et al., 2021 additionally uses the so-called experience replay technique with the target network, and also has a strong assumption on the boundedness of the feature matrix. Furthermore, the optimality is only guaranteed under a specific type of Markov Decision Process. Maei et al., 2010 suggested the so-called Greedy-GQ (gradient Q-learning) algorithm, but due to non-convexity of the objective function, it could converge to a local optima. Lu et al. (2021) used linear programming approach (Manne, 1960) to design convergent Q-learning algorithm under deterministic control systems. Devraj and Meyn (2017) proposed a Q-learning algorithm that minimizes asymptotic variance. However, it requires the assumption that number of changes of policy are finite, and involves matrix inversion at each iteration. Meyn (2023) introduced an optimistic training scheme with modified Gibbs policy for Q-learning with linear function approximation, which guarantees existence of a solution of the projected Bellman equation.

## 2 PRELIMINARIES AND NOTATIONS

### 2.1 MARKOV DECISION PROCESS

We consider an infinite horizon Markov Decision Process (MDP), which consists of a tuple $\mathcal{M} = (\mathcal{S}, \mathcal{A}, P, r, \gamma)$, where the state space $\mathcal{S}$ and action space $\mathcal{A}$ are finite sets, $P$ denotes the transition probability, $r : \mathcal{S} \times \mathcal{A} \times \mathcal{S} \to \mathbb{R}$ is the reward, and $\gamma \in (0, 1)$ is the discount factor.

Given a stochastic policy $\pi : \mathcal{S} \to \mathcal{P}(\mathcal{A})$, where $\mathcal{P}(\mathcal{A})$ is the set of probability distributions over $\mathcal{A}$, agent at the current state $s_k$ selects an action $a_k \sim \pi(\cdot|s_k)$, then the agent's state changes to the next state $s_{k+1} \sim P(\cdot|s_k, a_k)$, and receives reward $r_{k+1} := r(s_k, a_k, s_{k+1})$. A deterministic policy is a special stochastic policy, which can be defined simply as a mapping $\pi : \mathcal{S} \to \mathcal{A}$, which maps a state to an action.

The objective of MDP is to find a deterministic optimal policy, denoted by $\pi^*$, such that the cumulative discounted rewards over infinite time horizons is maximized, i.e., $\pi^* := \arg\max_\pi \mathbb{E}\left[\sum_{k=0}^{\infty} \gamma^k r_k \mid \pi\right]$, where $(s_0, a_0, s_1, a_1, \ldots)$ is a state-action trajectory generated by the Markov chain under policy $\pi$, and $\mathbb{E}[\cdot|\pi]$ is an expectation conditioned on the policy $\pi$. The Q-function under policy $\pi$ is defined as $Q^\pi(s, a) = \mathbb{E}\left[\sum_{k=0}^{\infty} \gamma^k r_k \mid s_0 = s, a_0 = a, \pi\right]$, $s \in \mathcal{S}$, $a \in \mathcal{A}$, and the optimal Q-function is defined as $Q^*(s, a) = Q^{\pi^*}(s, a)$ for all $s \in \mathcal{S}, a \in \mathcal{A}$. Once $Q^*$ is known, then an optimal policy can be retrieved by the greedy action, i.e., $\pi^*(s) = \arg\max_{a \in \mathcal{A}} Q^*(s, a)$. Throughout, we assume that the Markov chain is time homogeneous so that the MDP is well posed, which is standard in the literature.

It is known that the optimal Q-function satisfies the so-called Bellman equation expressed as follows:

$$Q^*(s, a) = \mathbb{E}\left[r_{k+1} + \max_{a_{k+1} \in \mathcal{A}} \gamma Q^*(s_{k+1}, a_{k+1}) \,\middle|\, s_k = s, a_k = a\right] := \mathcal{T}Q^*(s, a), \qquad (1)$$

where $\mathcal{T}$ is called the Bellman operator.

## 2.2 Notations

In this paper, we will use an O.D.E. model (Borkar and Meyn, 2000) of Q-learning to analyze its convergence. To this end, it is useful to introduce some notations in order to simplify the overall expressions. Throughout the paper, $e_a$ and $e_s$ denote $a$-th and $s$-th canonical basis vectors in $\mathbb{R}^{|\mathcal{A}|}$ and $\mathbb{R}^{|\mathcal{S}|}$, respectively. Moreover, $\otimes$ stands for the Kronecker product. Let us introduce the following notations:

$$P := \begin{bmatrix} P_1 \\ \vdots \\ P_{|\mathcal{A}|} \end{bmatrix} \in \mathbb{R}^{|\mathcal{S}||\mathcal{A}| \times |\mathcal{S}|}, \quad R := \begin{bmatrix} R_1 \\ \vdots \\ R_{|\mathcal{A}|} \end{bmatrix} \in \mathbb{R}^{|\mathcal{S}||\mathcal{A}|}, \quad Q := \begin{bmatrix} Q_1 \\ \vdots \\ Q_{|\mathcal{A}|} \end{bmatrix} \in \mathbb{R}^{|\mathcal{S}||\mathcal{A}|},$$

$$D_a := \begin{bmatrix} d(1, a) & & \\ & \ddots & \\ & & d(|\mathcal{S}|, a) \end{bmatrix} \in \mathbb{R}^{|\mathcal{S}| \times |\mathcal{S}|}, \quad D := \begin{bmatrix} D_1 & & \\ & \ddots & \\ & & D_{|\mathcal{A}|} \end{bmatrix} \in \mathbb{R}^{|\mathcal{S}||\mathcal{A}| \times |\mathcal{S}||\mathcal{A}|},$$

where $P_a \in \mathbb{R}^{|\mathcal{S}| \times |\mathcal{S}|}, a \in \mathcal{A}$ is the state transition matrix whose $i$-th row and $j$-th column component denotes the probability of transition to state $j$ when action $a$ is taken at state $i$, $P^\pi \in \mathbb{R}^{|\mathcal{S}||\mathcal{A}| \times |\mathcal{S}||\mathcal{A}|}$ represents the state-action transition matrix under policy $\pi$, i.e.,

$$(e_s \otimes e_a)^T P^\pi (e_{s'} \otimes e_{a'}) = \mathbb{P}[s_{k+1} = s', a_{k+1} = a'|s_k = s, a_k = a, \pi],$$

$Q_a = Q(\cdot, a) \in \mathbb{R}^{|\mathcal{S}|}, a \in \mathcal{A}$ and $R_a(s) := \mathbb{E}[r(s, a, s')|s, a], s \in \mathcal{S}$. Moreover, $d(\cdot, \cdot)$ is the state-action visit distribution, where i.i.d. random variables $\{(s_k, a_k)\}_{k=0}^{\infty}$ are sampled, i.e., $d(s, a) = \mathbb{P}[s_k = s, a_k = a]$, $a \in \mathcal{A}, s \in \mathcal{S}$. With a slight abuse of notation, $d$ will be also used to denote the vector $d \in \mathbb{R}^{|\mathcal{S}||\mathcal{A}|}$ such that $d^T(e_s \otimes e_a) = d(s, a)$, $\forall s \in \mathcal{S}, a \in \mathcal{A}$. In this paper, we represent a policy in a matrix form in order to formulate a switching system model. In particular, for a given policy $\pi$, define the matrix

$$\Pi^\pi := \begin{bmatrix} (e_{\pi(1)} \otimes e_1) & (e_{\pi(2)} \otimes e_2) & \cdots & (e_{\pi(|\mathcal{S}|)} \otimes e_{|\mathcal{S}|}) \end{bmatrix}^\top \in \mathbb{R}^{|\mathcal{S}| \times |\mathcal{S}||\mathcal{A}|}. \qquad (2)$$

Then, we can prove that for any deterministic policy, $\pi$, we have $\Pi^\pi Q = \begin{bmatrix} Q(1, \pi(1))^T & Q(2, \pi(2))^T & \cdots & Q(|\mathcal{S}|, \pi(|\mathcal{S}|))^T \end{bmatrix}^T$. For simplicity, let $\Pi_Q := \Pi^\pi$ when $\pi(s) = \arg\max_{a \in \mathcal{A}} Q(s, a)$. Moreover, we can prove that for any deterministic policy $\pi$, $P^\pi = P\Pi^\pi \in \mathbb{R}^{|\mathcal{S}||\mathcal{A}| \times |\mathcal{S}||\mathcal{A}|}$, where $P^\pi$ is the state-action transition probability matrix. Using the notations introduced, the Bellman equation in (1) can be compactly written as $Q^* = \gamma P\Pi_{Q^*} Q^* + R =: \mathcal{T}Q^*$, where $\pi_{Q^*}$ is the greedy policy defined as $\pi_{Q^*}(s) = \arg\max_{a \in \mathcal{A}} Q^*(s, a)$.

### 2.3 Q-LEARNING WITH LINEAR FUNCTION APPROXIMATION

Q-learning is widely used model-free learning to find $Q^*$, whose updates are given as

$$Q_{k+1}(s_k, a_k) \leftarrow Q_k(s_k, a_k) + \alpha_k \delta_k, \tag{3}$$

where $\delta_k = r_{k+1} + \gamma \max_{a \in \mathcal{A}} Q_k(s_{k+1}, a) - Q_k(s_k, a_k)$ is called the TD error. Each update uses an i.i.d. sample $(s_k, a_k, r_{k+1}, s_{k+1})$, where $(s_k, a_k)$ is sampled from a state-action distribution $d(\cdot, \cdot)$.

Here, we assume that the step-size is chosen to satisfy the so-called the Robbins-Monro condition (Robbins and Monro, 1951), $\alpha_k > 0$, $\sum_{k=0}^{\infty} \alpha_k = \infty$, $\sum_{k=0}^{\infty} \alpha_k^2 < \infty$. When the state-spaces and action-spaces are too large, then the memory and computational complexities usually become intractable. In such a case, function approximation is commonly used to approximate Q-function (Mnih et al., 2015; Schrittwieser et al., 2020; Hessel et al., 2018; Lan et al., 2020). Linear function approximation is one of the simplest function approximation approaches. In particular, we use the feature matrix $X \in \mathbb{R}^{|\mathcal{S}||\mathcal{A}| \times h}$ and parameter vector $\theta \in \mathbb{R}^h$ to approximate Q-function, i.e., $Q \simeq X\theta$, where the feature matrix is expressed as $X := \begin{bmatrix} x(1,1)^T & \cdots & x(1,|\mathcal{A}|)^T & \cdots & x(|\mathcal{S}|, |\mathcal{A}|)^T \end{bmatrix}^T \in \mathbb{R}^{|\mathcal{S}||\mathcal{A}| \times h}$. Here, $x(\cdot, \cdot) \in \mathbb{R}^h$ is called the feature vector, and $h$ is a positive integer with $h \ll |\mathcal{S}||\mathcal{A}|$. The corresponding greedy policy becomes $\pi_{X\theta}(s) = \arg\max_{a \in \mathcal{A}} x(s,a)^T \theta$. Note that the number of policies characterized by the greedy policy is finite. This is because the policy is invariant under constant multiplications, and there exists a finite number of sectors on which the policy is invariant. Next, we summarize some standard assumptions adapted throughout this paper.

**Assumption 2.1.** *The state-action visit distribution is positive, i.e., $d(s,a) > 0$ for all $s \in \mathcal{S}, a \in \mathcal{A}$.*

**Assumption 2.2.** *The feature matrix, $X$, has full column rank, and is a non-negative matrix. Moreover, columns of $X$ are orthogonal.*

**Assumption 2.3** (Boundedness on feature matrix and reward matrix)**.** *There exists constants, $X_{\max} > 0$ and $R_{\max} > 0$, such that $\max(||X||_\infty, ||X^T||_\infty) < X_{\max}$, $\quad ||R||_\infty < R_{\max}$.*

The Assumption 2.1, Assumption 2.2 and Assumption 2.3 are commonly adopted in the literature, e.g. Carvalho et al. (2020); Melo et al. (2008); Lee and He (2019). Moreover, under Assumption 2.1, $D$ is a nonsingular matrix with strictly positive diagonal elements.

**Lemma 2.4** (Gosavi (2006))**.** *Under Assumption 2.3, the optimal Q-function, $Q^*$, is bounded, i.e., $||Q^*||_\infty \leq \frac{R_{\max}}{1-\gamma}$.*

The proof of Lemma 2.4 comes from the fact that under the discounted infinite horizon setting, $Q^*$ can be expressed as an infinite sum of a geometric sequence.

**Remark 2.5.** *Carvalho et al., 2020; Agarwal et al., 2021 assume $||x(s,a)||_\infty \leq 1$ for all $(s,a) \in \mathcal{S} \times \mathcal{A}$. Moreover, Zhang et al. 2021 requires specific bounds on the feature matrix which is dependent on various factors e.g. projection radius and transition matrix . On the other hand, our feature matrix can be chosen arbitrarily large regardless of those factors.*

### 2.4 O.D.E. ANALYSIS

The dynamic system framework has been widely used to prove convergence of reinforcement learning algorithms, e.g., Sutton et al. 2009; Maei et al. 2010; Borkar and Meyn 2000; Lee and He 2019. Especially, Borkar and Meyn, 2000 is one of the most widely used techniques to prove stability of stochastic approximation using O.D.E. analysis. Consider the following stochastic algorithm with a nonlinear mapping $f : \mathbb{R}^n \to \mathbb{R}^n$:

$$\theta_{k+1} = f(\theta_k) + m_k, \tag{4}$$

where $m_k \in \mathbb{R}^n$ is an i.i.d. noise vector. For completeness, results in Borkar and Meyn, 2000 are briefly reviewed in the sequel. Under Assumption A.1 given in Appendix A.1, we now introduce Borkar and Meyn theorem below.

**Lemma 2.6** (Borkar and Meyn theorem)**.** *Suppose that Assumption A.1 in the Appendix A.1 holds, and consider the stochastic algorithm in (4). Then, for any initial $\theta_0 \in \mathbb{R}^n$, $\sup_{k \geq 0} ||\theta_k|| < \infty$ with probability one. In addition , $\theta_k \to \theta^e$ as $k \to \infty$ with probability one, where $f(\theta^e) = 0$.*

The main idea of Borkar and Meyn theorem is as follows: iterations of a stochastic recursive algorithm follow the solution of its corresponding O.D.E. in the limit when the step-size satisfies the Robbins-Monro condition. Hence, by proving the asymptotic stability of the O.D.E., we can induce the convergence of the original algorithm. In this paper, we will use an O.D.E. model of Q-learning, which is expressed as a special nonlinear system called a switching system. In the sequel, basic concepts in switching system theory are briefly introduced.

## 2.5 Switching System

In this paper, we will consider a particular nonlinear system, called the *switched linear system* (Liberzon, 2003),

$$\dot{x}_t = A_{\sigma_t} x_t, \quad x_0 = z \in \mathbb{R}^n, \quad t \in \mathbb{R}_+, \tag{5}$$

where $x_t \in \mathbb{R}^n$ is the state, $\mathcal{M} := \{1, 2, \ldots, M\}$ is called the set of modes, $\sigma_t \in \mathcal{M}$ is called the switching signal, and $\{A_\sigma, \sigma \in \mathcal{M}\}$ are called the subsystem matrices. The switching signal can be either arbitrary or controlled by the user under a certain switching policy. Especially, a state-feedback switching policy is denoted by $\sigma(x_t)$.

Stability and stabilization of (5) have been widely studied for decades. Still, finding a practical and effective condition for them is known to be a challenging open problem. Contrary to linear time-invariant systems, even if each subsystem matrix $A_\sigma$ is Hurwitz, the overall switching system may not be stable in general. This tells us that tools in linear system theories cannot be directly applied to conclude the stability of the switching system.

Another approach is to use the Lyapunov theory (Khalil, 2002). From standard results in control system theories, finding a Lyapunov function ensures stability of the switching system. If the switching system consists of negative definite matrices, we can always find a common quadratic Lyapunov function to ensure its stability. We use this fact to prove the convergence of the proposed algorithm. In particular, the proposed Q-learning algorithm can be modelled as a switching system, whose subsystem matrices are all negative definite.

## 3 Projected Bellman equation

In this section, we introduce the notion of projected Bellman equation with a regularization term, and establish connections between it and the proposed algorithm. Moreover, we briefly discuss the existence and uniqueness of the solution of the projected Bellman equation. We will also provide an example to illustrate the existence and uniqueness. When using the linear function approximation, since the true action value may not lie in the subspace spanned by the feature vectors, a solution of the Bellman equation may not exist in general. To resolve this issue, a standard approach is to consider the projected Bellman equation defined as

$$X\theta^* = \Gamma \mathcal{T} X \theta^*, \tag{6}$$

where $\Gamma := X(X^T D X)^{-1} X^T D$ is the weighted Euclidean Projection with respect to state-action visit distribution onto the subspace spanned by the feature vectors, and $\mathcal{T} X \theta^* = \gamma P \Pi_{X\theta^*} X \theta^* + R$. In this case, there is more chances for a solution satisfying the above projected Bellman equation to exist. Still, there may exist cases where the projected Bellman equation does not admit a solution. To proceed, let us rewrite (6) equivalently as

$$X\theta^* = X(X^T D X)^{-1} X^T D(\gamma P \Pi_{X\theta^*} X\theta^* + R) \Leftrightarrow \underbrace{(X^T D X - \gamma X^T D P \Pi_{X\theta^*} X)}_{A_{\pi_{X\theta^*}}} \theta^* = \underbrace{X^T D R}_{b},$$

where we use the simplified notations $A_{\pi_{X\theta^*}} := X^T D X - \gamma X^T D P \Pi_{X\theta^*} X$, $b = X^T D R$. Furthermore we use the simplified notation $C := X^T D X$. Therefore, the projected Bellman equation in (6) can be equivalently written as the nonlinear equation

$$b - A_{\pi_{X\theta^*}} \theta^* = 0. \tag{7}$$

A potential deterministic algorithm to solve the above equation is

$$\theta_{k+1} = \theta_k + \alpha_k(b - A_{\pi_{X\theta_k}}\theta_k). \tag{8}$$

If it converges, i.e., $\theta_k \to \theta^*$ as $k \to \infty$, then it is clear that $\theta^*$ solves (7). In this paper, the proposed algorithm is a stochastic algorithm that solves the modified equation

$$b - (A_{\pi_{X\theta_e}} + \eta I)\theta_e = 0, \tag{9}$$

where $I$ is the $|\mathcal{S}||\mathcal{A}| \times |\mathcal{S}||\mathcal{A}|$ identity matrix, and $\eta \geq 0$ is a weight on the regularization term. We can use $\eta C$ instead of $\eta I$ as the regularization term but $\eta C$ is known to solve a MDP with modified discount factor Chen et al. (2022). Similar to (8), the corresponding deterministic algorithm is

$$\theta_{k+1} = \theta_k + \alpha_k(b - (A_{\pi_{X\theta_k}} + \eta I)\theta_k). \tag{10}$$

If it converges, i.e., $\theta_k \to \theta_e$ as $k \to \infty$, then it is clear that $\theta_e$ solves (9). Some natural questions that arise here are as follows: Which conditions can determine the existence and uniqueness of the equations in (7 and 9)? Partial answers are given in the sequel. Considering the non-existence of fixed point of (6) (De Farias and Van Roy, 2000), both 7 and (9) may not also have a solution. However, for the modified Bellman equation in (9), we can prove that under appropriate conditions, its solution exists and is unique. We give an example where the solution does not exist for (7) but does exist for (9) in Appendix A.10.

**Lemma 3.1.** *When $\eta > X_{\max}^2\sqrt{|\mathcal{S}||\mathcal{A}|} - \lambda_{\min}(C)$, a solution of (9) exists and is unique.*

The proof is given in Appendix A.3, which uses Banach fixed-point theorem (Agarwal et al., 2018). From Lemma 3.1, we can see that when the weight $\eta$ is sufficiently large, the existence and uniqueness of the solution is guaranteed. Note that even if a solution satisfying (9) exists, $X\theta_e$ may be different from the optimal Q-function, $Q^*$. However, we can derive a bound on the error, $X\theta_e - Q^*$, using some algebraic inequalities and contraction property of Bellman operator, which is presented below.

**Lemma 3.2.** *Assume that a solution of (9) exists. When $\eta > X_{\max}^2\sqrt{|\mathcal{S}||\mathcal{A}|} - \lambda_{\min}(C)$, we have the following bound:*

$$||X\theta_e - Q^*||_\infty \leq \frac{\eta X_{\max}^2|\mathcal{S}||\mathcal{A}|}{\lambda_{\min}(C)(\eta + \lambda_{\min}(C) - X_{\max}^2\sqrt{|\mathcal{S}||\mathcal{A}|})} \frac{2R_{\max}}{1-\gamma}$$
$$+ \frac{\lambda_{\min}(C) + \eta}{\eta + \lambda_{\min}(C) - X_{\max}^2\sqrt{|\mathcal{S}||\mathcal{A}|}} ||\Gamma Q^* - Q^*||_\infty.$$

Some remarks are in order for Lemma 3.2. First of all, $\eta > X_{\max}^2\sqrt{|\mathcal{S}||\mathcal{A}|} - \lambda_{\min}(C)$ ensures that the error is always bounded. The first term represents the error potentially induced by the regularization. The second term represents the error incurred by the difference between the optimal $Q^*$ and $Q^*$ projected onto the feature space. Therefore, this error is induced by the linear function approximation. Note that even if $\eta \to \infty$, the error remains bounded.

## 4 ALGORITHM

In this section, we will introduce our main algorithm, called RegQ, and elaborate the condition on the regularization term to make the algorithm convergent. The proposed algorithm is motivated by TD-learning. In particular, for on-policy TD-learning, one can establish its convergence using the property of the stationary distribution. On the other hand, for an off-policy case, the mismatch between the sampling distribution and the stationary distribution could cause its divergence (Sutton et al., 2016). To address this problem, Diddigi et al., 2019 adds a regularization term to TD-learning in order to make it convergent. Since Q-learning can be interpreted as an off-policy TD-learning, we add a regularization term to Q-learning update motivated by Diddigi et al., 2019. This modification leads to the proposed RegQ algorithm as follows:

$$\theta_{k+1} = \theta_k + \alpha_k(x(s_k, a_k)\delta_k + \eta\theta_k) \tag{11}$$

The pseudo-code is given in Appendix A.9. Note that letting $\eta = 0$, the above update is reduced to the standard Q-learning with linear function approximation in (3). The proposed RegQ is different from Diddigi et al., 2019 in the sense that a regularization term is applied to Q-learning instead of TD-learning. Rewriting the stochastic update in a deterministic manner, it can be written as follows:

$$\theta_{k+1} = \theta_k + \alpha_k(b - (A_{\pi_{X\theta_k}} + \eta I)\theta_k + m_{k+1}), \tag{12}$$

where $m_{k+1} = \delta_k x(s_k, a_k) + \eta\theta_k - (b - (A_{\pi_{X\theta_k}} + \eta I)\theta_k)$ is an i.i.d. noise. Note that without the noise, (12) is reduced to the deterministic version in (10). In our convergence analysis, we will apply the O.D.E. approach, and in this case, $A_{\pi_{X\theta_k}} + \eta I$ will determine the stability of the corresponding O.D.E. model, and hence, convergence of (11). Note that (12) can be interpreted as a switching system defined in (5) with stochastic noises. As mentioned earlier, proving the stability of a general switching system is challenging in general. However, we can find a common Lyapunov function to prove its asymptotic stability. In particular, we can make $-(A_{\pi_{X\theta_k}} + \eta I)$ to be negative definite under the following condition:

$$\eta > \lambda_{\max}(C)\left(\max_{\pi \in \Theta, s \in \mathcal{S}, a \in \mathcal{A}} \frac{\gamma d^T P^\pi(e_a \otimes e_s)}{2d(s,a)} - \frac{2-\gamma}{2}\right), \tag{13}$$

where $\Theta$ is the set of all deterministic policies, and $\otimes$ is the Kronecker product. Lemma A.3, given in Appendix A.2, is similar to Theorem 2 in Diddigi et al., 2019, and ensures such a property. Now, we can use the Lyapunov argument to establish stability of the overall system. Building on the negative definiteness of the $-(A_{\pi_{X\theta_k}} + \eta I)$, in the next section, we prove that under the stochastic update (11), we have $\theta_k \to \theta_e$ as $k \to \infty$ with probability one, where $\theta_e$ satisfies the projected Bellman equation in (9). If $\eta = 0$ satisfies (13), we can guarantee convergence to an optimal policy without errors.

## 5  CONVERGENCE ANALYSIS

Recently, Lee and He, 2019 suggested a switching system framework to prove the stability of Q-learning in the linear function approximation cases. However, its assumption seems too stringent to check in practice. Here, we develop more practical Q-learning algorithm by adding an appropriately preconditioned regularization term. We prove the convergence of the proposed Q-learning with regularization term (11) following lines similar to Lee and He, 2019. Our proof mainly relies on Borkar-Meyn theorem. Therefore, we first discuss about the corresponding O.D.E. for the proposed update in (11), which is

$$\dot{\theta}_t = -(X^T D X + \eta I)\theta_t + \gamma X^T D P \Pi_{X\theta_t} X \theta_t + X^T D R := f(\theta_t). \tag{14}$$

Then, using changes of coordinates, the above O.D.E. can be rewritten as

$$\frac{d}{dt}(\theta_t - \theta_e) = (-X^T D X - \eta I + \gamma X^T D P \Pi_{X\theta_t} X)(\theta_t - \theta_e) + \gamma X^T D P(\Pi_{X\theta_t} - \Pi_{X\theta_e})X\theta_e, \tag{15}$$

where $\theta_e$ satisfies (9). Here, we assume that the equilibrium point exists and is unique. We later prove that if the equilibrium exists, then it is unique. To apply Borkar-Meyn theorem, we discuss about the asymptotic stability of the O.D.E. in (15), and check conditions of Assumption A.1 in Appendix A.1. Note that (15) includes an affine term, i.e., it cannot be expressed as a matrix times $\theta_t - \theta_e$. Establishing asymptotic stability of switched linear system with affine term is difficult compared to switched linear system (5). To circumvent this difficulty, Lee and He, 2019 proposed upper and lower systems, which upper bounds and lower bounds the original system, respectively using the so-called vector comparison principle. Then, the stability of the original system can be established by proving the stability of the upper and lower systems, which are easier to analyze. Following similar lines, to check global asymptotic stability of the original system, we also introduce upper and lower systems, which upper bounds and lower bounds the original system, respectively. Then, we prove global asymptotic stability of the two bounding systems. Since upper and lower systems can be viewed as switched linear system and linear system, respectively, the global asymptotic stability is easier to prove. We stress that although the switching system

approach in Lee and He, 2019 is applied in this paper, the detailed proof is entirely different and nontrivial. In particular, the upper and lower systems are given as follows:

$$\dot{\theta}_t^u = (-X^T D X - \eta I + \gamma X^T D P \Pi_{X\theta_t^u} X)\theta_t^u, \quad \dot{\theta}_t^l = -(X^T D X - \eta I + \gamma X^T D P \Pi_{X\theta_e} X)\theta_t^l,$$

where $\theta_t^u$ denotes the state of the upper system, and $\theta_t^l$ stands for the state of the lower system. We defer the detailed construction of each system to Appendix A.6. Establishing stability of upper and lower system gives the stability of overall system.

**Theorem 5.1.** *Suppose that (a) Assumption 2.2 holds, (b) (13) holds, and (c) a solution of (9) exists. Then, the solution is unique, and the origin is the unique globally asymptotically stable equilibrium point of (15).*

The detailed proof is given in Appendix A.6.

Building on previous results, we now use Borkar and Meyn's theorem in Lemma 2.6 to establish the convergence of RegQ. The full proof of the following theorem is given in Appendix A.7.

**Theorem 5.2.** *If $\eta$ satisfies (13), then with Assumption 2.1, Assumption 2.2 and Assumption 2.3, under the stochastic update (11), $\theta_k \to \theta_e$ as $k \to \infty$ with probability one, where $\theta_e$ satisfies (9).*

## 6 Experiments

In this section, we present experimental results under well-known environments in Tsitsiklis and Van Roy (1996); Baird (1995), where Q-learning with linear function approximation diverges. In Appendix A.8.3, we also compare performance under the Mountain Car environment (Sutton and Barto, 2018) where Q-learning performs well. In Appendix A.8.2, we show experimental results under various step-size and $\eta$. We also show trajectories of upper and lower systems to illustrate the theoretical results.

### 6.1 $\theta \to 2\theta$ (Tsitsiklis and Van Roy, 1996)

Even when there are only two states, Q-learning with linear function approximation could diverge (Tsitsiklis and Van Roy, 1996). Depicted in Figure 3a in Appendix A.8.1, from state one ($\theta$), the transition is deterministic to absorbing state two ($2\theta$), and reward is zero at every time steps. Therefore, the episode length is fixed to be two. Learning rate for Greedy GQ (GGQ) and Coupled Q Learning (CQL), which have two learning rates, are set as 0.05 and 0.25, respectively as in Carvalho et al., 2020; Maei et al., 2010. Since CQL requires normalized feature values, we scaled the feature value with $\frac{1}{2}$ as in Carvalho et al., 2020, and initialized weights as one. We implemented Q-learning with target network (Zhang et al., 2021), which also have two learning rates, without projection for practical reason (Qtarget). We set the learning rate as 0.25 and 0.05 respectively, and the weight $\eta$ as two. For RegQ, we set the learning rate as 0.25, and the weight $\eta$ as two. It is averaged over 50 runs. In Figure 1a, we can see that RegQ achieves the fastest convergence rate.

### 6.2 Baird Seven Star Counter Example (Baird, 1995)

Baird, 1995 considers an overparameterized example, where Q-learning with linear function approximation diverges. The overall state transition is depicted in Figure 3b given in Appendix A.8.1. There are seven states and two actions for each state, which are solid and dash action. The number of features are $h = 15$. At each episode, it is initialized at random state with uniform probability. Solid action leads to seventh state while dashed action makes transition uniformly random to states other than seventh state. At seventh state, the episode ends with probability $\frac{1}{100}$. The behavior policy selects dashed action with probability $\frac{5}{6}$, and solid action with probability $\frac{1}{6}$. Since CQL in Carvalho et al. (2020) converges under normalized feature values, we scaled the feature matrix with $\frac{1}{\sqrt{5}}$. The weights are set as one except for $\theta_7 = 2$. The learning rates and the weight $\eta$ are set as same as the previous experiment. As in Figure 1b, Our RegQ shows the fastest convergence compared to other convergent algorithms.

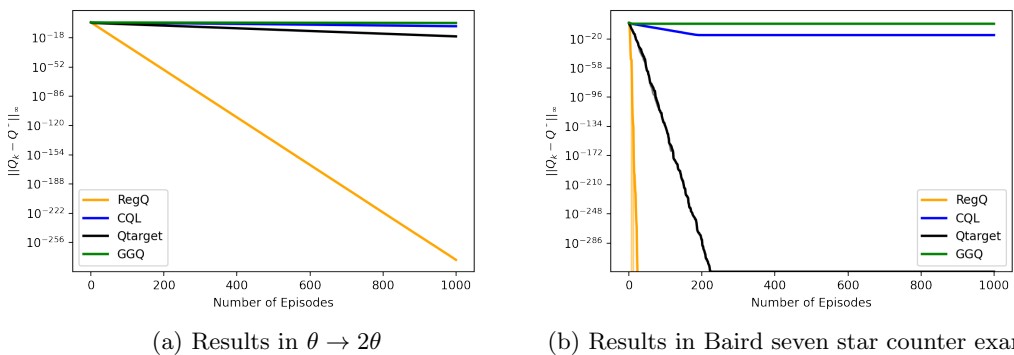

(a) Results in $\theta \to 2\theta$          (b) Results in Baird seven star counter example

Figure 1: Experiment results

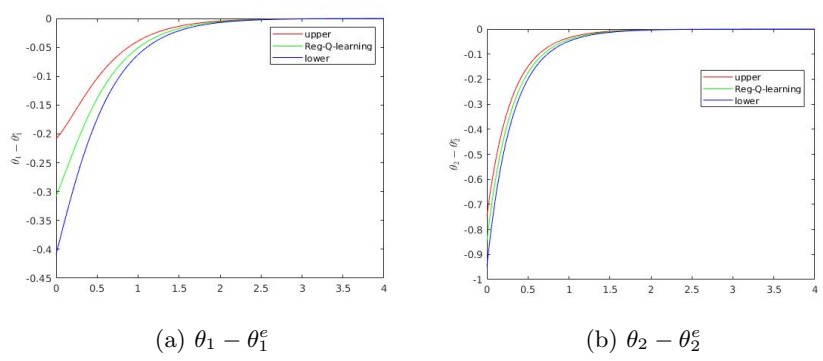

(a) $\theta_1 - \theta_1^e$          (b) $\theta_2 - \theta_2^e$

Figure 2: O.D.E. results

## 6.3 O.D.E. EXPERIMENT

Let us consider a MDP with $|\mathcal{S}| = 2, |\mathcal{A}| = 2$, and the following parameters:

$$X = \begin{bmatrix} 1 & 0 \\ 0 & 2 \\ 1 & 0 \\ 0 & 2 \end{bmatrix}, \quad D = \begin{bmatrix} \frac{1}{4} & 0 & 0 & 0 \\ 0 & \frac{1}{4} & 0 & 0 \\ 0 & 0 & \frac{1}{4} & 0 \\ 0 & 0 & 0 & \frac{1}{4} \end{bmatrix}, \quad P = \begin{bmatrix} 0.5 & 0.5 \\ 1 & 0 \\ 0.5 & 0.5 \\ 0.25 & 0.75 \end{bmatrix}, \quad R = \begin{bmatrix} 1 \\ 1 \\ 1 \\ 1 \end{bmatrix}, \quad \gamma = 0.99.$$

For this MDP, we will illustrate trajectories of the upper and lower system. Each state action pair is sampled uniformly random and reward is one for every time step. $\eta = 2.25$ is chosen to satisfy conditions of Theorem 5.1. From Figure 2, we can see that the trajectory of the original system is bounded by the trajectories of lower and upper system.

## 7 CONCLUSION

In this paper, we presented a new convergent Q-learning with linear function approximation (RegQ), which is simple to implement. We provided theoretical analysis on the proposed RegQ, and demonstrated its performance on several experiments, where the original Q-learning with linear function approximation diverges. Developing a new Q-learning algorithm with linear function approximation without bias would be one interesting future research topic. Moreover, considering the great success of deep learning, it would be interesting to develop deep reinforcement learning algorithms with appropriately preconditioned regularization term instead of using the target network.

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

# A    APPENDIX

## A.1    ASSUMPTION FOR BORKAR AND MEYN THEOREM

**Assumption A.1.**
1. The mapping $f : \mathbb{R}^n \to \mathbb{R}^n$ is globally Lipschitz continuous, and there exists a function $f_\infty : \mathbb{R}^n \to \mathbb{R}^n$ such that

$$\lim_{c \to \infty} \frac{f(cx)}{c} = f_\infty(x), \quad \forall x \in \mathbb{R}^n. \tag{16}$$

2. The origin in $\mathbb{R}^n$ is an asymptotically stable equilibrium for the O.D.E. $\dot{x}_t = f_\infty(x_t)$.

3. There exists a unique globally asymptotically stable equilibrium $\theta^e \in \mathbb{R}^n$ for the O.D.E. $\dot{x}_t = f(x_t)$ , i.e., $x_t \to \theta^e$ as $t \to \infty$.

4. The sequence $\{m, k \geq 1\}$ where $\mathcal{G}_k$ is sigma-alebra generated by $\{(\theta_i, m_i, i \geq k)\}$, is a Martingale difference sequence. In addition , there exists a constant $C_0 < \infty$ such that for any initial $\theta_0 \in \mathbb{R}^n$ , we have $\mathbb{E}[||m_{k+1}||^2 |\mathcal{G}_k] \leq C_0(1 + ||\theta_k||^2), \forall k \geq 0$.

5. The step-sizes satisfies the Robbins-Monro condition (Robbins and Monro, 1951) :

$$\sum_{k=0}^{\infty} \alpha_k = \infty, \quad \sum_{k=0}^{\infty} \alpha_k^2 < \infty.$$

## A.2    POSITIVE DEFINITENESS OF $A_{\pi_{X\theta}} + \eta I$

We first introduce Gerschgorin circle theorem (Horn and Johnson, 2013) to prove Lemma A.3.

**Lemma A.2** (Gerschgorin circle theorem (Horn and Johnson, 2013)). *Let $A = [a_{ij}] \in \mathbb{R}^{n \times m}$ and $R_i(A) = \sum\limits_{j \neq i}^{m} a_{ij}$. Consider the Gerschgorin circles*

$$\{z \in \mathbb{C}| : |z - a_{ii}| \leq R_i(A)\}, \quad i = 1, \ldots, n.$$

*The eigenvalues of $A$ are in the union of Gerschgorin discs*

$$G(A) = \cup_{i=1}^{n} \{z \in \mathbb{C}| : |z - a_{ii}| \leq R_i(A)\}.$$

Now, we state the lemma to guarantee positive definiteness of $A_{\pi_{X\theta}} + \eta I$. Instead we prove positive definiteness of $A_{\pi_{X\theta}} + \frac{\eta}{\lambda_{\max}(C)} C$.

**Lemma A.3.** *Let*

$$M^{\pi_{X\theta}} := D\left(\left(1 + \frac{\eta}{\lambda_{\max}(C)}\right) I - \gamma P^{\pi_{X\theta}}\right).$$

*Under the following condition:*

$$\eta > \lambda_{\max}(C) \max_{\pi \in \Theta, s \in \mathcal{S}, a \in \mathcal{A}} \left(\frac{\gamma d^T P^{\pi_{X\theta}} e_i}{2d_i} - \frac{2 - \gamma}{2}\right) \quad ,$$

*where $\Theta$ is the set of all deterministic policies, and $\otimes$ is the Kronecker product, $M_{\pi_{X\theta}}$ is positive definite.*

*Proof.* We use Gerschgorin circle theorem for the proof. First, denote $m_{ij} = [M_{\pi_{X\theta}}]_{ij}$. Then, one gets

$$m_{ii} = d_i \left(\left(1 + \frac{\eta}{\lambda_{\max}(C)}\right) - \gamma e_i^T P^{\pi_{X\theta}} e_i\right),$$
$$m_{ij} = -d_i \gamma e_i^T P^{\pi_{X\theta}} e_j \quad \text{for} \quad i \neq j.$$

Except for the diagonal element, the row and column sums, respectively, become

$$\sum_{j \in S_i} |m_{ij}| = \gamma d_i (1 - e_i^T P^{\pi X \theta} e_i)$$

$$\sum_{j \in S_i} |m_{ji}| = \gamma d^T P^{\pi X \theta} e_i - \gamma d_i e_i^T P^{\pi X \theta} e_i,$$

where $S_i = \{1, 2, \ldots, |\mathcal{S}||\mathcal{A}|\} \setminus \{i\}$. We need to show that $M^{\pi X \theta} + M^{\pi X \theta^T}$ is positive definite. To this end, we use Lemma A.2 to have the following inequality:

$$|\lambda - 2m_{ii}| \leq \sum_{j \in S_i} |m_{ij}| + \sum_{j \in S_i} |m_{ji}|$$

Considering the lower bound of $\lambda$, we have

$$\lambda \geq 2m_{ii} - \sum_{j \in S_i} |m_{ij}| - \sum_{j \in S_i} |m_{ji}|$$

$$= 2d_i \left( \left( 1 + \frac{\eta}{\lambda_{\max}(C)} \right) - \gamma e_i^T P^{\pi X \theta} e_i \right) - \gamma d_i (1 - e_i^T P^{\pi X \theta} e_i) - (\gamma d^T P^{\pi X \theta} e_i - \gamma d_i e_i^T P^{\pi X \theta} e_i)$$

$$= \eta \frac{2d_i}{\lambda_{\max}(C)} + (2 - \gamma) d_i - \gamma d^T P^{\pi X \theta} e_i.$$

Hence, for $\lambda > 0$, we should have

$$\eta > \lambda_{\max}(C) \left( \frac{\gamma d^T P^{\pi X \theta} e_i}{2d_i} - \frac{2 - \gamma}{2} \right).$$

Taking $\eta > \lambda_{\max}(C) \max_{\pi \in \Theta, s \in \mathcal{S}, a \in \mathcal{A}} \left( \frac{\gamma d^T P^{\pi X \theta} e_i}{2d_i} - \frac{2-\gamma}{2} \right)$, we can make $M^{\pi X \theta}$ always positive definite. $\qquad \square$

### A.3 Proof of Lemma 3.1

To show existence and uniqueness of the solution of (9), we use Banach fixed-point theorem (Agarwal et al., 2018). First, we define the operator $\mathcal{T}_\eta$ as follows:

$$\mathcal{T}_\eta(\theta) := (X^T D X + \eta I)^{-1} (X^T D R + \gamma X^T D P \Pi_{X\theta} X \theta)$$

We show that $\mathcal{T}_\eta$ is contraction mapping. The existence and uniqueness of (9) follows from the Banach fixed-point theorem.

$$\begin{aligned}
||\theta_1 - \theta_2||_\infty &= ||(X^T D X + \eta I)^{-1} (\gamma X^T D P \Pi_{X\theta_1} X \theta_1 - \gamma X^T D P \Pi_{X\theta_2} X \theta_2)||_\infty \\
&\leq \gamma ||(X^T D X + \eta I)^{-1}||_\infty ||X^T||_\infty ||\Pi_{X\theta_1} X \theta_1 - \Pi_{\pi X\theta_2} X \theta_2||_\infty \\
&\leq \gamma ||(X^T D X + \eta I)^{-1}||_\infty ||X^T||_\infty ||\Pi_{X(\theta_1 - \theta_2)} (X \theta_1 - X \theta_2)||_\infty \\
&\leq \gamma ||(X^T D X + \eta I)^{-1}||_\infty ||X^T||_\infty ||X \theta_1 - X \theta_2||_\infty \\
&\leq \gamma ||(X^T D X + \eta I)^{-1}||_\infty ||X^T||_\infty ||X||_\infty ||\theta_1 - \theta_2||_\infty \\
&\leq \gamma \frac{\sqrt{|\mathcal{S}||\mathcal{A}|}}{\lambda_{\min}(C) + \eta} ||X^T||_\infty ||X||_\infty ||\theta_1 - \theta_2||_\infty \\
&\leq \gamma ||\theta_1 - \theta_2||_\infty.
\end{aligned}$$

The first inequality follows from the sub-multiplicativity of matrix norm and $||DP||_\infty \leq 1$. The second inequality follows from the fact that $\max x - \max y \leq \max(x - y)$. The last inequality is due to the condition $\eta > X_{\max}^2 \sqrt{|\mathcal{S}||\mathcal{A}|} - \lambda_{\min}(C)$. Since $\gamma < 1$, $\mathcal{T}_\eta$ is contraction mapping. Now we can use Banach fixed-point theorem to conclude existence and uniqueness of (9).

## A.4 Proof of Lemma 3.2

*Proof.* Let $\Gamma_\eta := X(X^T D X + \eta I)^{-1} X^T D$. The bias term of the solution can be obtained using simple algebraic inequalities.

$$
\begin{aligned}
||X\theta_e - Q^*||_\infty &= ||\Gamma_\eta \mathcal{T}(X\theta_e) - \Gamma\mathcal{T}(Q^*)||_\infty + ||\Gamma\mathcal{T}Q^* - Q^*||_\infty \\
&\leq ||\Gamma_\eta \mathcal{T}(X\theta_e) - \Gamma_\eta \mathcal{T}(Q^*)||_\infty + ||\Gamma_\eta \mathcal{T}(Q^*) - \Gamma\mathcal{T}(Q^*)||_\infty + ||\Gamma Q^* - Q^*||_\infty \\
&\leq \gamma ||\Gamma_\eta||_\infty ||X\theta_e - Q^*||_\infty + ||\Gamma_\eta - \Gamma||_\infty ||\mathcal{T}(Q^*)||_\infty + ||\Gamma Q^* - Q^*||_\infty
\end{aligned}
$$

The last inequality follows from the fact that the Bellman operator $\mathcal{T}$ is $\gamma$-contraction with respect to the infinity norm. We bound each terms. First, we have

$$
\begin{aligned}
||\Gamma_\eta||_\infty &= \left\|X(X^T D X + \eta I)^{-1} X^T D\right\|_\infty \\
&\leq X_{\max}^2 \left\|(X^T D X + \eta I)^{-1}\right\|_\infty \\
&\leq X_{\max}^2 \sqrt{|\mathcal{S}||\mathcal{A}|} \left\|(X^T D X + \eta I)^{-1}\right\|_2 \\
&\leq \frac{X_{\max}^2 \sqrt{|\mathcal{S}||\mathcal{A}|}}{\lambda_{\min}(C) + \eta}.
\end{aligned}
$$

The first inequality follows from Assumption 2.3. The second inequality follows from the matrix norm inequality and the last inequality follows from the fact that induced norm of symmetric positive definite matrix equals its maximum eigenvalue.

Bounding $||\Gamma_\eta - \Gamma||_\infty$, we have

$$
\begin{aligned}
||\Gamma_\eta - \Gamma||_\infty &\leq ||X||_\infty ||X^T||_\infty ||(X^T D X + \eta I)^{-1} - (X^T D X)^{-1}||_\infty \\
&\leq X_{\max}^2 \left\|(X^T D X + \eta I)^{-1}(X^T D X - (X^T D X + \eta I))(X^T D X)^{-1}\right\|_\infty \\
&\leq \eta X_{\max}^2 \left\|(X^T D X + \eta I)^{-1}\right\|_\infty \left\|(X^T D X)^{-1}\right\|_\infty \\
&\leq \frac{\eta X_{\max}^2 |\mathcal{S}||\mathcal{A}|}{\lambda_{\min}(C)(\lambda_{\min}(C) + \eta)}.
\end{aligned}
$$

Lastly, we have $||\mathcal{T}(Q^*)||_\infty \leq \frac{2R_{\max}}{1-\gamma}$ which follows from boundedness of $Q^*$ in Lemma 2.4.

Arranging the terms, we get

$$
\frac{\eta + \lambda_{\min}(C) - X_{\max}^2 \sqrt{|\mathcal{S}||\mathcal{A}|}}{\lambda_{\min}(C) + \eta}||X\theta_e - Q^*||_\infty \leq \frac{\eta X_{\max}^2 |\mathcal{S}||\mathcal{A}|}{\lambda_{\min}(C)(\lambda_{\min}(C) + \eta)} \frac{2R_{\max}}{1-\gamma} + ||\Gamma Q^* - Q^*||_\infty
$$

Rearranging the terms, we have

$$
||X\theta_e - Q^*||_\infty \leq \frac{\eta X_{\max}^2 |\mathcal{S}||\mathcal{A}|}{\lambda_{\min}(C)(\eta + \lambda_{\min}(C) - X_{\max}^2 \sqrt{|\mathcal{S}||\mathcal{A}|})} \frac{2R_{\max}}{1-\gamma} + \frac{\lambda_{\min}(C) + \eta}{\eta + \lambda_{\min}(C) - X_{\max}^2 \sqrt{|\mathcal{S}||\mathcal{A}|}} ||\Gamma Q^* - Q^*||_\infty.
$$

The bias is caused by projection and additional error term due to regularization. □

## A.5 Proofs to check Assumption A.1 for Theorem 5.2.

In this section, we provide omitted proofs to check Assumption A.1 for Theorem 5.2.

First of all, Lipschitzness of $f(\theta)$ ensures the unique solution of the O.D.E..

**Lemma A.4** (Lipschitzness). *Let*

$$
f(\theta) = -(X^T D X + \eta I)\theta + \gamma X^T D P \Pi_{X\theta} X\theta + X^T D R. \tag{17}
$$

*Then, $f(\theta)$ is globally Lipschitzness continuous.*

*Proof.* Lipschitzness of $f(\theta)$ can be proven as follows:

$$
\begin{aligned}
||f(\theta) - f(\theta')||_\infty &\leq ||(X^T D X + \eta I)(\theta - \theta')||_\infty + \gamma ||X^T D P (\Pi_{X\theta} X\theta - \Pi_{X\theta'} X\theta')||_\infty \\
&\leq ||X^T D X + \eta I||_\infty ||\theta - \theta'||_\infty + \gamma ||X^T D P||_\infty ||\Pi_{X\theta} X\theta - \Pi_{X\theta'} X\theta'||_\infty \\
&\leq ||X^T D X + \eta I||_\infty ||\theta - \theta'||_\infty + \gamma ||X^T D P||_\infty ||\Pi_{X(\theta-\theta')} X(\theta - \theta')||_\infty \\
&\leq (||X^T D X + \eta I||_\infty + \gamma ||X^T D P||_\infty ||X||_\infty)||\theta - \theta'||_\infty
\end{aligned}
$$

Therefore $f(\theta)$ is Lipschitz continuous with respect to the $||\cdot||_\infty$, □

Next, the existence of limiting O.D.E. of (14) can be proved using the fact that policy is invariant under constant multiplication when linear function approximation is used.

**Lemma A.5** (Existence of limiting O.D.E. and stability). *Let*

$$f(\theta) = (-X^T D X - \eta I)\theta + \gamma X^T D P \Pi_{X\theta} X \theta + X^T D R. \tag{18}$$

*Under (13), there exists limiting O.D.E. of (18) and its origin is asymptotically stable.*

*Proof.* The existence of limiting O.D.E. can be obtained using the homogeneity of policy, $\Pi_{X(c\theta)} = \Pi_{X\theta}$.

$$f(c\theta) = -(X^T D X + \eta I)(c\theta) + \gamma X^T D P \Pi_{X(c\theta)} X (c\theta) + X^T D R,$$

$$\lim_{c \to \infty} \frac{f(cx)}{c} = (-X^T D X - \eta I + \gamma X^T D P \Pi_{X\theta} X)\theta$$

This can be seen as switching system and shares common Lyapunov function $V = ||\theta||^2$. Hence, the origin is asymptotically stable. $\qquad\square$

Lastly, we check conditions for martingale difference sequences.

**Lemma A.6** (Martingale difference sequence, $m_k$, and square integrability). *We have*

$$\mathbb{E}[m_{k+1}|\mathcal{F}_k] = 0,$$
$$\mathbb{E}[||m_{k+1}||^2|\mathcal{F}_k] < C_0(1 + ||\theta||^2),$$

*where $C_0 := \max(12X_{\max}^2 R_{\max}^2, 12\gamma X_{\max}^4 + 4\eta^2)$.*

*Proof.* To show $\{m_k, k \in \mathbb{N}\}$ is a martingale difference sequence with respect to the sigma-algebra generated by $\mathcal{G}_k$, we first prove expectation of $m_{k+1}$ is zero conditioned on $\mathcal{G}_k$:

$$\mathbb{E}[m_{k+1}|\mathcal{G}_k] = 0$$

This follows from definition of $b, C$ and $A_{\pi_{X\theta}}$.

The boundedness $\mathbb{E}[||n_k||] < \infty$ also follows from simple algebraic inequalities. Therefore $\{m_k, k \in \mathbb{N}\}$ is martingale difference sequence.

Now, we show square integrability of the martingale difference sequence, which is

$$\mathbb{E}[||m_{k+1}||^2|\mathcal{G}_k] \leq C_0(||\theta_k||^2 + 1).$$

Using simple algebraic inequalities, we have

$$\begin{aligned}
\mathbb{E}[||m_{k+1}||^2|\mathcal{G}_k] &= \mathbb{E}[||\delta_k x(s_k, a_k) + \eta\theta_k - \mathbb{E}_\mu[\delta_k x(s_k, a_k) + \eta\theta_k]||^2|\mathcal{G}_t] \\
&\leq \mathbb{E}[||\delta_k x(s_k, a_k) + \eta\theta_k||^2 + ||\mathbb{E}_\mu[\delta_k x(s_k, a_k) + \eta\theta_k]||^2|\mathcal{G}_t] \\
&\leq 2\mathbb{E}[||\delta_k x(s_k, a_k) + \eta\theta_k||^2\mathcal{G}_t] \\
&\leq 4\mathbb{E}[||\delta_k x(s_k, a_k)||^2|\mathcal{G}_t] + 4\eta^2\mathbb{E}[||\theta_k||^2|\mathcal{G}_t] \\
&\leq 12X_{\max}^2\mathbb{E}[||r_k||^2 + ||\gamma \max x(s_k, a_k)\theta_k||^2 + ||x(s_k, a_k)\theta_k||^2|\mathcal{G}_t] + 4\eta^2||\theta_k||^2 \\
&\leq 12X_{\max}^2 R_{\max}^2 + 12\gamma X_{\max}^4||\theta_k||^2 + ||\theta_k||^2 + 4\eta^2||\theta_k||^2 \\
&\leq C_0(1 + ||\theta_k||^2),
\end{aligned}$$

where $C_0 := \max(12X_{\max}^2 R_{\max}^2, 12\gamma X_{\max}^4 + 4\eta^2)$. The fourth inequality follows from the fact that $||a + b + c||^2 \leq 3||a||^2 + 3||b||^2 + 3||c||^2$. This completes the proof.

$\qquad\square$

A.6  Proof of Theorem 5.1

Before moving onto the proof of Theorem 5.1, in order to prove the stability using the upper and lower systems, we need to introduce some notions such as the quasi-monotone function and vector comparison principle. We first introduce the notion of quasi-monotone increasing function, which is a necessary prerequisite for the comparison principle for multidimensional vector system.

**Definition A.7** (Quasi-monotone function). *A vector-valued function $f : \mathbb{R}^n \to \mathbb{R}^n$ with $f := [f_1 \quad f_2 \quad \cdots \quad f_n]^T$ is said to be quasi-monotone increasing if $f_i(x) \leq f_i(y)$ holds for all $i \in \{1, 2, \ldots, n\}$ and $x, y \in \mathbb{R}^n$ such that $x_i = y_i$ and $x_j \leq y_j$ for all $j \neq i$.*

Based on the notion of quasi-monotone function, we introduce the vector comparison principle.

**Lemma A.8** (Vector Comparison Principle (Hirsch and Smith, 2006)). *Suppose that $\bar{f}, \underline{f}$ are globally Lipschitz continuous. Let $x_t$ be a solution of the system*

$$\frac{d}{dt}x_t = \bar{f}(x_t), \qquad x_o \in \mathbb{R}^n, \forall t \geq 0.$$

*Assume that $\bar{f}$ is quasi-monotone increasing, and let $v_t$ be a solution of the system*

$$\frac{d}{dt}v_t = \underline{f}(v_t), \qquad v_0 < x_0, \forall t \geq 0,$$

*where $\underline{f}(v) \leq \bar{f}(v)$ holds for any $v \in \mathbb{R}^n$. Then, $v_t \leq x_t$ for all $t \geq 0$.*

The vector comparison lemma can be used to bound the state trajectory of the original system by those of the upper and lower systems. Then, proving global asymptotic stability of the upper and lower systems leads to global asymptotic stability of original system. We now give the proof of Theorem 5.1.

*Proof.* First we construct the upper comparison part. Noting that

$$\gamma X^T DP\Pi_{X\theta_e}X\theta_e \geq \gamma X^T DP\Pi_{X\theta}X\theta_e \tag{19}$$

and

$$\gamma X^T DP\Pi_{X(\theta-\theta_e)}X(\theta - \theta_e) \geq \gamma X^T DP\Pi_{X\theta}X(\theta - \theta_e), \tag{20}$$

we define $\bar{f}(y)$ and $\underline{f}(y)$ as follows:

$$\bar{f}(y) = (-X^T DX - \eta I + \gamma X^T DP\Pi_{Xy}X)y,$$

$$\underline{f}(y) = (-X^T DX - \eta I + \gamma X^T DP\Pi_{X(y+\theta_e)}X)y + \gamma X^T DP(\Pi_{X(y+\theta_e)} - \Pi_{X\theta_e})X\theta_e$$

Using (19) and (20), we have $\bar{f}(y) \leq \underline{f}(y)$.

$\underline{f}$ is the corresponding O.D.E. of original system and $\bar{f}$ becomes O.D.E. of the upper system. $\bar{f}$ becomes switched linear system.

Now consider the O.D.E. systems

$$\frac{d}{dt}\theta_t^u = \bar{f}(\theta_t^u), \qquad \theta_0^u > \theta_0,$$

$$\frac{d}{dt}\theta_t = \underline{f}(\theta_t).$$

Next, we prove quasi-monotone increasing property of $\bar{f}$. For any $z \in \mathbb{R}^{|S||A|}$, consider a non-negative vector $p \in \mathbb{R}^{|S||A|}$ such that its $i$-th element is zero. Then, for any $1 \leq i \leq d$, we have

$$\begin{aligned}
e_i^T \bar{f}(y + p) &= e_i^T(-X^T DX - \eta I + \gamma X^T DP\Pi_{X(y+p)}X)(y + p) \\
&= -e_i^T(X^T DX + \eta I)y - \eta e_i^T p + \gamma e_i^T X^T DP\Pi_{X(y+p)}X(y + p) \\
&\geq -e_i^T(X^T DX + \eta I)y + \gamma e_i^T X^T DP\Pi_{Xy}Xy \\
&= e_i^T \bar{f}(y),
\end{aligned}$$

where the inequality comes from $e_i^T X^T D X p = 0$ due to Assumption 2.2 and $e_i^T p = 0$ since $i$-th element of $p$ is zero.

Therefore by Lemma A.8, we can conclude that $\theta_t \leq \theta_t^u$. The switching system matrices of the upper system are all negative definite by Lemma A.3, The switching system shares $V(\theta) = ||\theta||^2$ as common Lyapunov function. Therefore, we can conclude that the upper comparison system is globally asymptotically stable.

For the lower comparison part, noting that

$$\gamma X^T D P \Pi_{X\theta} X\theta \geq \gamma X^T D P \Pi_{X\theta_e} X\theta,$$

we can define $\underline{f}(y)$ and $\bar{f}(y)$ such that $\underline{f}(y) \leq \bar{f}(y)$ as follows:

$$\bar{f}(y) = -X^T D X y - \eta y + \gamma X^T D P \Pi_{Xy} X y + X^T D R,$$
$$\underline{f}(y) = -X^T D X y - \eta y + \gamma X^T D P \Pi_{X\theta_e} X y + X^T D R$$

The corresponding O.D.E. system becomes

$$\frac{d}{dt}\theta_t = \bar{f}(\theta_t),$$
$$\frac{d}{dt}\theta_t^l = \underline{f}(\theta_t^l), \quad \theta_0^l < \theta_0. \tag{21}$$

Proving quasi-monotonicity of $\bar{f}$ is similar to previous step. Consider non-negative vector $p \in \mathbb{R}^{|S||A|}$ such that its $i$-th element is zero. Then, we have

$$\begin{aligned}
e_i^T \bar{f}(y+p) &= e_i^T(-(X^T D X + \eta I)(y+p) + \gamma X^T D P \Pi_{X(y+p)} X(y+p) + X^T D R) \\
&= e_i^T(-(X^T D X + \eta I)y + \gamma X^T D P \Pi_{X(y+p)} X(y+p) + X^T D R) \\
&\geq e_i^T(-(X^T D X + \eta I)y + \gamma X^T D P \Pi_{Xy} X y + X^T D R) \\
&= e_i^T \bar{f}(y).
\end{aligned}$$

The second equality holds since $X^T D X$ is diagonal matrix and $p_i = 0$.
Therefore by Lemma A.8, we can conclude that $\theta_t^l \leq \theta_t$. The lower comparison part is linear system with affine term, and the matrix is negative definite by Lemma A.3. Hence, we can conclude that (21) is globally asymptotically stable.

To prove uniqueness of the equilibrium point, assume there exists two different equilibrium points $\theta_1^e$ and $\theta_2^e$. The global asymptotic stability implies that regardless of initial state, $\theta_t \to \theta_1^e$ and $\theta_t \to \theta_2^e$. However this becomes contradiction if $\theta_1^e \neq \theta_2^e$. Therefore, the equilibrium point is unique. □

### A.7 Proof of Theorem 5.2

*Proof.* To apply Lemma 2.6, let us check Assumption A.1.

1. First and second statement of Assumption A.1 follows from Lemma A.5

2. Third statement of Assumption A.1 follows from Theorem 5.1

3. Fourth statement of Assumption A.1 follows from Lemma A.6

Since we assumed Robbins Monro step-size, we can now apply Lemma 2.6 to complete the proof. □

### A.8 Experiments

#### A.8.1 Diagrams for $\theta \to 2\theta$ and Baird Seven Star Counter Example

The state transition diagrams of $\theta \to 2\theta$ and Baird seven-star example are depicted in Figure 3a and Figure 3b respectively.

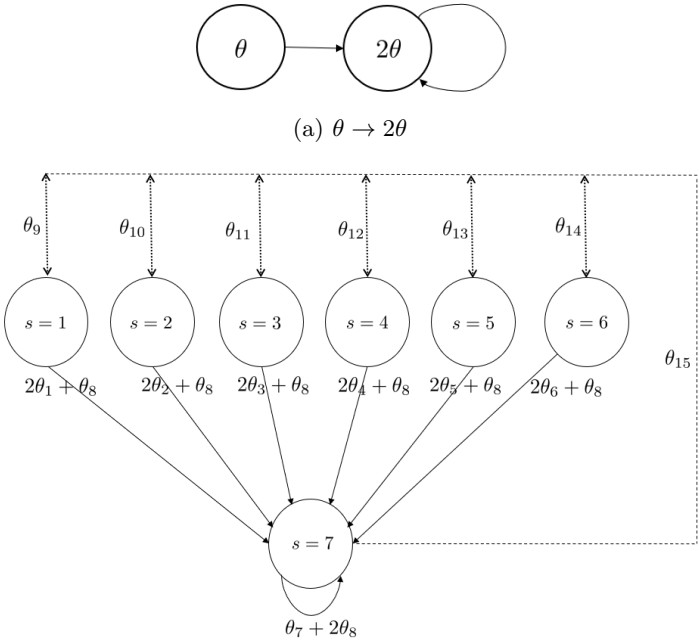

(a) $\theta \to 2\theta$

(b) Baird seven star counter example

Figure 3: Counter-examples where Q-learning with linear function approximation diverges

A.8.2 Experiments with varying hyperparameters

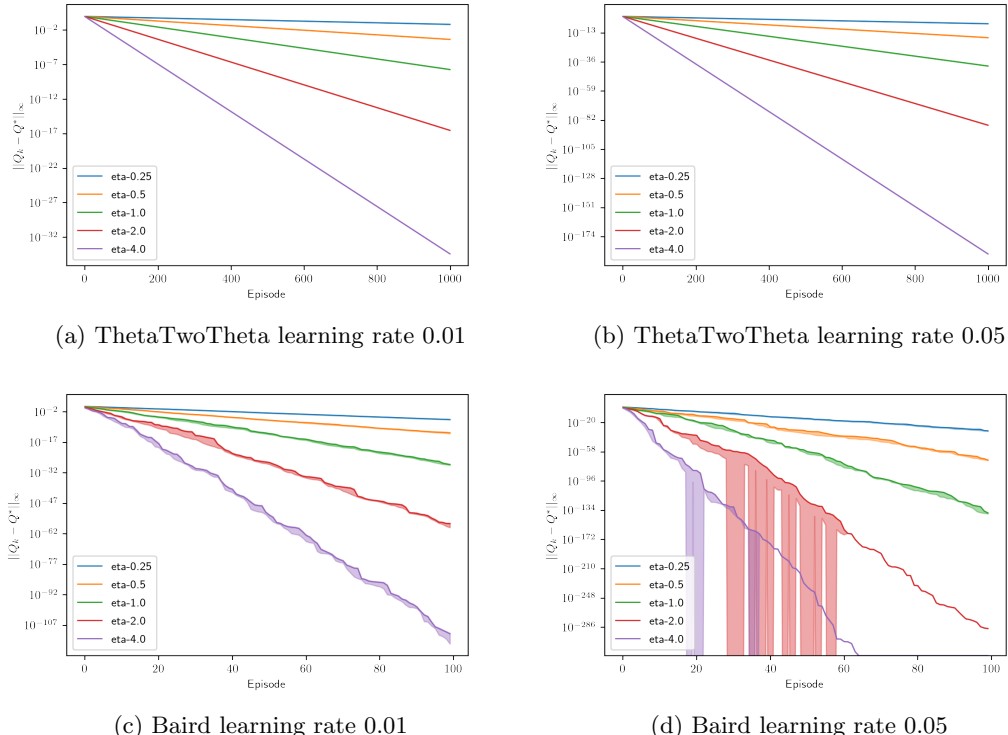

(a) ThetaTwoTheta learning rate 0.01

(b) ThetaTwoTheta learning rate 0.05

(c) Baird learning rate 0.01

(d) Baird learning rate 0.05

Figure 4: Learning curve under different learning rate and regularization coefficient

In Figure 4, we have ran experiments under $\eta \in \{2^{-2}, 2^{-1}, 1, 2, 4\}$, and learning rate $0..01, 0.05$. Overall, we can see that the convergence rate gets faster as $\eta$ increases.

### A.8.3 MOUNTAIN CAR (SUTTON AND BARTO, 2018) EXPERIMENT

Mountain Car is environment where state consists of position, and velocity, which are both continuous values. The actions are discrete, accelerating to left, staying neutral, and accelerating to the right. The goal is to reach the top of the mountain quickly as agent gets -1 reward every time step. We use tile-coding (Sutton and Barto, 2018) to discretize the states. We experimented under various tiling numbers and with appropriate $\eta$, it achieves performance as Q-learning does. We ran 1000 episodes for the training process, and the episode reward was averaged for 100 runs during test time. From Table 1, with appropriate $\eta$, RegQ performs comparable to Q-learning.

Table 1: Result of episode reward, step size $= 0.1$. The columns correspond to $\eta$, and rows correspond to number of tiles.

|  | 0 | 0.01 | 0.05 | 0.1 |
|---|---|---|---|---|
| $2 \times 2$ | $-199.993 \pm 0.005$ | $-200.0 \pm 0.0$ | $\mathbf{-199.28 \pm 0.074}$ | $-199.993 \pm 0.005$ |
| $4 \times 4$ | $-196.631 \pm 0.179$ | $\mathbf{-189.903 \pm 0.225}$ | $-194.178 \pm 0.166$ | $-196.631 \pm 0.179$ |
| $8 \times 8$ | $-185.673 \pm 0.305$ | $\mathbf{-163.08 \pm 0.248}$ | $-185.103 \pm 0.219$ | $-185.673 \pm 0.305$ |
| $16 \times 16$ | $-166.893 \pm 0.33$ | $\mathbf{-158.152 \pm 0.251}$ | $-167.934 \pm 0.238$ | $-166.893 \pm 0.33$ |

## A.9 Pseudo-code

---
**Algorithm 1** Regularized Q-learning
---
1: Initialize $\theta_0 \in \mathbb{R}^n$.
2: Set the step-size $(\alpha_k)_{k=0}^{\infty}$, and the behavior policy $\mu$.
3: **for** iteration $k = 0, 1, \ldots$ **do**
4:      Sample $s_k \sim d^{\mu}$ and $a_k \sim \mu$.
5:      Sample $s'_k \sim P(s_k, a_k, \cdot)$ and $r_{k+1} = r(s_k, a_k, s'_k)$.
6:      Update $\theta_k$ using (11).
7: **end for**
---

## A.10 Example of discussion on solution of the Projected Bellman equation

We now provide an example where the solution does not exist for (7) but does exist for (9).

**Example A.9.** *Let us define a MDP whose state transition diagram is given as in Figure 5. The cardinality of state space and action space are $|\mathcal{S}| = 3$, $|\mathcal{A}| = 2$ respectively. The*

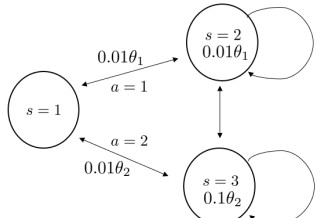

Figure 5: State transition diagram

*corresponding state transition matrix, and other parameters are given as follows:*

$$X = \begin{bmatrix} 0.01 & 0 \\ 0.1 & 0 \\ 0 & 0.01 \\ 0 & 0.01 \\ 0.1 & 0 \\ 0 & 0.01 \end{bmatrix}, \ R_1 = \begin{bmatrix} -2 \\ 0 \\ 0 \end{bmatrix}, \ R_2 = \begin{bmatrix} 1 \\ 0 \\ 0 \end{bmatrix}, \ P_1 = \begin{bmatrix} 0 & 1 & 0 \\ \frac{1}{4} & \frac{1}{4} & \frac{1}{2} \\ \frac{1}{4} & \frac{1}{2} & \frac{1}{4} \end{bmatrix}, \ P_2 = \begin{bmatrix} 0 & 0 & 1 \\ \frac{1}{4} & \frac{1}{4} & \frac{1}{2} \\ \frac{1}{4} & \frac{1}{2} & \frac{1}{4} \end{bmatrix},$$

$$\gamma = 0.99, \quad d(s, a) = \frac{1}{6}, \ \forall s \in \mathcal{S}, \forall a \in \mathcal{A}$$

*where the order of elements of each column follows the orders of the corresponding definitions. Note that for this Markov decision process, taking action $a = 1$ and action $a = 2$ at state $s = 2$ have the same transition probabilities and reward. It is similar for the state $s = 3$. In this MDP, there are only two deterministic policies available, denoted by $\pi_1$ and $\pi_2$, that selects action $a = 1$ and action $a = 2$ at state $s = 1$, respectively, i.e., $\pi_1(1) = 1$ and $\pi_2(1) = 2$. The actions at state $s = 2$ and $s = 3$ do not affect the overall results.*

The motivation of this MDP is as follows. Substitute $\pi_{X\theta^*}$ in (7) with $\pi_1$ and $\pi_2$. Then each of its solution becomes

$$\theta^{e1} := \begin{bmatrix} \theta_1^{e1} \\ \theta_2^{e1} \end{bmatrix} \approx \begin{bmatrix} 6 \\ 111 \end{bmatrix} \in \mathbb{R}^2, \quad \theta^{e2} := \begin{bmatrix} \theta_1^{e2} \\ \theta_2^{e2} \end{bmatrix} \approx \begin{bmatrix} -496 \\ -4715 \end{bmatrix} \in \mathbb{R}^2.$$

If $\pi_1$ is the corresponding policy to the solution of (7), it means that action $a = 1$ is greedily selected at state $s = 1$. Therefore, $Q^{\pi_1}(1, 1) > Q^{\pi_1}(1, 2)$ should be satisfied. However, since $Q^{\pi_1}(1, 1) = x(1, 1)^T \theta^{e1} = 0.06$ and $Q^{\pi_1}(1, 2) = x(1, 2)^T \theta^{e1} \approx 1.11$, this is contradiction. The same logic applies to the case for $\pi_2$. Therefore, neither of them becomes a solution of (7). On the other hand, considering (9) with $\eta = 2$ which satisfies (13), the solution for each policy becomes $\theta_1^{e1} \approx -1.66 \cdot 10^{-5}, \theta_2^{e1} \approx 8.3 \cdot 10^{-6}$ and $\theta_1^{e2} \approx -0.0016, \theta_2^{e2} \approx 0.00083$ respectively. For $\pi_1$ and $\pi_2$, we have $Q^{\pi_1}(1, 1) < Q^{\pi_2}(1, 2)$ and $Q^{\pi_1}(1, 1) < Q^{\pi_1}(1, 2)$ respectively. Hence, $\theta^{e2}$ satisfies (9) and becomes the unique solution.

