# OpenReview forum: "RegQ: Convergent Q-Learning with Linear Function Approximation using Regularization"
_ICLR.cc/2024/Conference — Submitted to ICLR 2024_

### Official Review · Reviewer_iu9a · 2023-10-31

**Soundness:** 3 good
**Presentation:** 3 good
**Contribution:** 2 fair
**Rating:** 5
**Confidence:** 3

**Summary:**

This paper introduces RegQ algorithm for Q-learning with linear function approximation. They introduced a regularization term in the Q-learning update (based on project bellman equation), which renders existence of solution to the modified target equation. They then proved boundedness (stability) and convergence of  their proposed algorithm. They also demonstrated their algorithm converges faster on some examples (the Baird star example).

**Strengths:**

quality: They used linear switching system theory to prove stability of the proposed RegQ algorithm (Thm 5.1), while many convergence proofs of Q-learning algorithms need to assume stability.

**Weaknesses:**

novelty and significance:
1. Adding regularization term to linear system to obtain/enhance solution is a common idea. I believe the real question is the performance of the resulting solution in terms of cumulative rewards or policy performance, which is ultimately what we care about. This paper only shows convergence of the algorithm to some solution, but the quality of that target solution resulting from regularized projected Bellman equation is not analyzed.

related work:
The paper missed some recent results on Q-learning with linear function approximation. S. Meyn showed existence of solution to projected Bellman equation with linear function approximation (and a stable algorithm) under assumptions of behavior policy.

ref:
S. Meyn, Stability of Q-Learning Through Design and Optimism, https://arxiv.org/abs/2307.02632, 2023

**Questions:**

NA

---

> ### Author Response · Authors · 2023-11-20
> **Response to Reviewer iu9a**
>
> We thank the reviewer for the valuable feedback. We have provided detailed responses to the reviewer's questions and comments.
>
> **1. Adding regularization term to linear system to obtain/enhance solution is a common idea. I believe the real question is the performance of the resulting solution in terms of cumulative rewards or policy performance, which is ultimately what we care about. This paper only shows convergence of the algorithm to some solution, but the quality of that target solution resulting from regularized projected Bellman equation is not analyzed.**
>
> We would like to note that an error bound from the original solution is provided in Lemma 3.2. Moreover, performance was evaluated in three different domains in Section 6, and RegQ shows better or comparable results compared to existing algorithms.
>
> Moreover, although adding a regularization term to linear system to enhance solution is a common idea, this idea was not fully studied for switching systems to the authors' knowledge. The main technical breakthrough in this paper is that a common quadratic Lyapunov function is found for the underlying switching system model, and it is used for the convergence analysis.
>
> **2. related work: The paper missed some recent results on Q-learning with linear function approximation. S. Meyn showed existence of solution to projected Bellman equation with linear function approximation (and a stable algorithm) under assumptions of behavior policy.**
>
> We agree with the reviewer's comment, and thank you for introducing the promising recent work of S. Meyn, where the existence of the solution is shown under "Tamed Gibbs policy" which is a version of softmax policy. Following the reviewer's comment, Meyn's work has been newly added in the revised version, and the related work part has been modified accordingly.

---

### Official Review · Reviewer_qe9J · 2023-10-31

**Soundness:** 2 fair
**Presentation:** 3 good
**Contribution:** 1 poor
**Rating:** 5
**Confidence:** 3

**Summary:**

The paper focuses on the Q-learning algorithm solving the problem of finite-state finite action MDP with linear approximation. Although the convergence of the lookup table based Q-learning is well-established, the linear approximation based Q-learning is known to be unstable for certain cases/examples. This paper proposes a new Q-learning algorithm by adding an additional regularization term to stabilize the learning process and proves that it will converge to a unique solution asymptotically. The author/s also show the proposed algorithm's effectiveness in solving the examples that previous Q-learning algorithm usually fails to converge.

**Strengths:**

1. The new proposed Q-learning algorithm is proven to be stable and asymptotically converge to a unique solution compared to the previous Q-learning methods in linear approximation setup.
2. The paper also gives a bound on the error between the estimated action value function and the optimal one, which consists of two terms: the error originated from the added regularization term and the error from the difference between the optimal Q value function and the one after feature vector projection.

**Weaknesses:**

1. Although it's good to have a new algorithm in linear approximation Q-learning to converge to a unique solution asymptotically, the added regularization term increases the error bound between the estimated action value function and the optimal one. Based on the result in Lemma 3.2, it's not clear how the \eta parameter should be chosen. A relatively small \eta could potentially have very small denominator in the RHS, causing big error bound. Instead, if the \eta is chosen super big, the error tends to remain the same. The experiment in the Appendix also shows the larger \eta helps with the convergence rate. Does that mean we should always use a super big \eta?
2. The added regularization + asymptotic analysis seems not requiring that much effort given the previous works on Q-learning about O.D.E. analysis, switching system, and off-policy TD-learning.

**Questions:**

Based on the experiment in the Appendix, larger \eta seems to be helping increase convergence rate and reduce the estimation error. Does that mean we should always go with larger \eta value?

---

> ### Author Response · Authors · 2023-11-20
> **Response to Reviewer qe9J**
>
> We thank the reviewer for the valuable feedback. We have provided detailed responses to the reviewer's questions and comments.
>
> **1. Although it's good to have a new algorithm in linear approximation Q-learning to converge to a unique solution asymptotically, the added regularization term increases the error bound between the estimated action value function and the optimal one. Based on the result in Lemma 3.2, it's not clear how the $\eta$ parameter should be chosen. A relatively small $\eta$ could potentially have very small denominator in the RHS, causing big error bound. Instead, if the $\eta$ is chosen super big, the error tends to remain the same. The experiment in the Appendix also shows the larger $\eta$ helps with the convergence rate. Does that mean we should always use a super big $\eta$?**
>
> Too large $\eta$ will cause high variance in the update because the algorithm is implemented in a stochastic sense. In general, the choice of $\eta$ leads to a trade-off between the error and convergence property, and one needs to find a compromise between the two extremes, which may need further studies. In practice, we may choose decreasing $\eta$, i.e., $\eta_k = \frac{\eta_0}{k+1}$ with some positive constant $\eta_0$ to mitigate the bias coming from the regularization coefficient.
>
>
> **2. The added regularization + asymptotic analysis seems not requiring that much effort given the previous works on Q-learning about O.D.E. analysis, switching system, and off-policy TD-learning.**
>
>
> Compared to the existing off-policy TD-learning analysis, the greedy policy of Q-learning, which is time-varying, makes the analysis much more intricate.
>
> Compared to the recently developed switching system-based analysis of Q-learning, adding regularization term to the system in our work requires more careful analysis. In particular, the main breakthrough in the current work is that we found a common quadratic Lyapunov function and used it for convergence analysis using a special structure of the switching system. This new finding is significant because it is in general hard to find a common quadratic Lyapunov function for switching systems. We expect that this new insight can be also used for Q-learning to accelerate its convergence speed, which can be potential future topics.
>
> **3. Based on the experiment in the Appendix, larger $\eta$ seems to be helping increase convergence rate and reduce the estimation error. Does that mean we should always go with larger $\eta$ value?**
>
>
> As can be seen from the experiment in the Mountain car, large $\eta$ does not necessarily imply better performance.
> In general, the regularization coefficient $\eta$ provides a trade-off between the convergence property and the bias error.
> In particular, larger $\eta$ leads to (faster) convergence while increased error. On the other hand, with small $\eta$, the proposed regQ becomes more similar to the vanilla Q-learning, and hence, it can fail to converge, while the error may be reduced.
>
> Compared to the Mountain car, the solution of $\theta\to 2\theta$ and Baird environment are the origin. Hence, adding regularization term does not incur any bias and improves the convergence rate.

---

### Official Review · Reviewer_aR4a · 2023-11-01

**Soundness:** 3 good
**Presentation:** 3 good
**Contribution:** 3 good
**Rating:** 6
**Confidence:** 4

**Summary:**

This paper proposes a new algorithm called regQ for reinforcement learning. The existence and uniqueness of the associated stochastic approximation problem is proved. By leveraging recent ODE-based analysis, the authors show the convergence of the updating rule. Numerical experiments are conducted to evaluate the algorithm performance.

**Strengths:**

1. The paper is overall well-written and easy to follow.
2. The paper proposes a new algorithm, and the authors analyze the convergence.
3. Numerical experiments are provided.

**Weaknesses:**

1. Seemingly, the novelty is an additional regularization term in the TD error to ensure the existence and uniqueness of the solution. It is unclear whether this new algorithm is better than vanilla Q-learning.
2. Although convergence of the regQ is provided, the paper does not show whether one can learn the optimal state-action value function $Q^*$ well. Indeed, we do not know whether the $Q$-value associated with $\eta > 0$ converges to the optimal $Q^*$ of the MDP problem when $\eta$ goes to zero.
3. The experiment setting cannot reflect the practical need. More comprehensive comparison with benchmarks is necessary.

**Questions:**

1. It is not clear whether linear function approximation makes analysis much harder. What are the technical difficulties in the convergence analysis?
2. Do you have a sample complexity analysis of regQ? For Q-learning algorithms, both asymptotic and non-asymptotic analyses are well-understood.

---

> ### Author Response · Authors · 2023-11-20
> **Response to Reviewer aR4a**
>
> We thank the reviewer for the valuable feedback. We have provided detailed responses to the reviewer's questions and comments.
>
> **1. Seemingly, the novelty is an additional regularization term in the TD error to ensure the existence and uniqueness of the solution. It is unclear whether this new algorithm is better than vanilla Q-learning.**
>
> We highlight several points that show that the proposed RegQ is better then vanilla Q-learning :
>
>  First, as for vanilla Q-learning, when linear function approximation is used together, one cannot guarantee its convergence, whereas convergence of our algorithm can be guaranteed theoretically, which is given in Theorem 5.2.
>
>  Secondly, as for vanilla Q-learning, there are cases where fixed points of the projected Bellman equation does not exist, and the corresponding example is given in Example A.9 in the Appendix. However, as for our modified projected Bellman equation, solutions always exist, and the corresponding example is given in Lemma 3.1
>
>  Lastly, our experiments included in the paper demonstrate that the proposed algorithm shows better or comparable results than vanilla Q-learning.
>
>
> **2. Although convergence of the regQ is provided, the paper does not show whether one can learn the optimal state-action value function well. Indeed, we do not know whether the Q-value associated with converges to the optimal of the MDP problem when goes to zero.**
>
> The iterates of the proposed algorithm is proved to converge to the optimal Q-function with some biases which can be controlled by the regularization coefficient. Moreover, an error bound with the optimal solution is given in Lemma~3.2. We note that to remedy the bias, we can use a diminishing regularization coefficient, i.e., $\eta=\frac{\eta_0}{1+k}$ at $k$-th iteration.
>
>
> **3. The experiment setting cannot reflect the practical need. More comprehensive comparison with benchmarks is necessary.**
>
>  We will consider adding more comprehensive comparisons with benchmarks. However, we want to emphasize that validating on simple environments is also important as sanity check and for providing further insights.
>
> **4. It is not clear whether linear function approximation makes analysis much harder. What are the technical difficulties in the convergence analysis?**
>
> Unlike the tabular setting, the linear function approximation case does not guarantee convergence for Q-learning, which is the main difficulty. This challenge can be overcome through the switched system analysis and introduction of a regularization coefficient.
>
> **5. Do you have a sample complexity analysis of regQ? For Q-learning algorithms, both asymptotic and non-asymptotic analyses are well-understood.**
>
> In this work, we did not provide sample complexity result for regQ. However, we believe the sample complexity result can be also derived for regQ following the recent technique in the literature of non-asymptotic analysis.

---

> > ### Comment · Reviewer_aR4a · 2023-11-21
> > **Response to authors**
> >
> > I want to thank the authors for their revision and response. My questions are well-addressed. It would be great if non-aymptotic could be derived in the future. However, the current version is already good. I have raised my score.

---

> > > ### Author Response · Authors · 2023-11-23
> > > **Response to Reviewer aR4a**
> > >
> > > We are glad to hear that most of the concerns have been addressed. We will consider non-asymptotic analysis in the future. We thank again the reviewer for the time and effort in reviewing the manuscript.

---

### Official Review · Reviewer_drQD · 2023-11-01

**Soundness:** 4 excellent
**Presentation:** 2 fair
**Contribution:** 3 good
**Rating:** 8
**Confidence:** 4

**Summary:**

This paper focuses on the problem of instability of $Q$-learning with linear function approximation. The paper shows that the ridge regularization of the parameters is enough to guarantee convergence of $Q$-learning. The quality of the solution is the upper-bounded.

**Strengths:**

This paper shows that simply regularizing the $Q$-learning with linear function approximation update by penalizing the weights in the 2-norm is enough to guarantee convergence. Even though other works had hinted at this insight, specifically Zhang~2021, in this work the contribution is distilled, in the sense that previous work had not only the ridge regularization but also other additions to the update.

The paper is clearly written and easy to follow. The switching systems technique used is also less common than just using the o.d.e. analysis, which makes the paper possibly technically more interesting.

**Weaknesses:**

While there is an upper bound on the quality of the regularized solution, it is important to understand if the regularization can make $Q$-learning useful in non-trivial environments. The experiments only show convergence to the correct solution in trivial environments. Nevertheless, this experimental validation could be seen as future work.

The work of Chen 2022 as hinted that regularizing the $Q$-learning objective can sometimes be seen as lowering the discount factor. If this is the case, it is uninspiring because low discount factors are known to lead to the convergence of $Q$-learning. These insights are not discussed in the paper.

Minor: the related work discussion in the introduction is confusing. Comparison with related work appears on the second, third, fourth and fifth paragraph, and some times it is repetitive. I think the paper would benefit from using the introduction for motivation and context and having a separate section for discussion with related work.

**Questions:**

- do the authors believe the method will be useful in practice, or that the regularization introduced will harm the quality too much? Have the authors performed more experiments to understand this?

---

> ### Author Response · Authors · 2023-11-20
> **Response to Reviewer drQD**
>
> We thank the reviewer for the valuable feedback. We have provided detailed responses to the reviewer's questions and comments.
>
> **1. While there is an upper bound on the quality of the regularized solution, it is important to understand if the regularization can make Q-learning useful in non-trivial environments. The experiments only show convergence to the correct solution in trivial environments. Nevertheless, this experimental validation could be seen as future work.**
>
> We agree with the reviewer that the algorithm should be tested on more complex environments. As for clarification, the experiment in  mountain car environment shows the performance rather than the convergence to the correct solution. Moreover, we will add related experiments in the future work.
>
>
> **2. The work of Chen 2022 as hinted that regularizing the
> Q-learning objective can sometimes be seen as lowering the discount factor. If this is the case, it is uninspiring because low discount factors are known to lead to the convergence of
> Q-learning. These insights are not discussed in the paper.**
>
>
> From Chen 2022, when using tabular setup, regularization has the effect of lowering the discount factor. However, when using linear function approximation, choosing $\eta I$ as a regularization, following the same logic in Chen 2022 does not lead to low discounted setting. We have marked the discussion in the manuscript with colored fonts.
>
>
> **3. Minor: the related work discussion in the introduction is confusing. Comparison with related work appears on the second, third, fourth and fifth paragraph, and some times it is repetitive. I think the paper would benefit from using the introduction for motivation and context and having a separate section for discussion with related work.**
>
> Following the reviewer's comment, in the revised version, we have polished the paragraphs and introduced a separate section for discussion with the related works.
>
>
> **4. Do the authors believe the method will be useful in practice, or that the regularization introduced will harm the quality too much? Have the authors performed more experiments to understand this?**
>
> We believe that the method will be useful in practice because many RL algorithms are known to be unstable. Moreover, in mountain-car environment, which is still a toy example, but close to practice, the suggested algorithm shows better or comparable results to the original Q-learning. Furthermore, we can also use a diminishing regularization coefficient, i.e., $\eta=\frac{\eta_0}{1+k}$ at $k$-th iteration to mitigate the effect of the regularization term for some positive constant $\eta_0$.

---

### Author Response · Authors · 2023-11-20
**Official Comment by Authors**

Dear all reviewers, we appreciate the fruitful and constructive comments to improve the manuscript. The changes in the revised manuscript are marked with ${\color{red} colored}$ fonts.

---

### Meta-Review · Area_Chair_pGEj · 2023-12-10

**Metareview:**

RegQ: Convergent Q-Learning with Linear Function Approximation using Regularization

This paper introduces RegQ, a novel Q-learning algorithm that guarantees convergence with linear function approximation. By incorporating a regularization term, RegQ achieves stability, as proven by a recent analysis using switching system models.

The reviews are split. The negative reviews are short and uninformative, where additional references and detailed clarification can be/have been addressed. On the other hand, I am concerned about the potential issue of how the regularization term changed the original problem into a different one (that bypasses the impossibility result). A positive reviewer (drQD) pointed out that adding the regularization term changed the original objective and the corresponding solution, which in an implicit way lowers the discount factor, as proved by Chen et al, 2022 for the tabular setting. The authors responded that the linear case is not as straightforward and discussed in detail. I checked the corresponding paragraphs in the current paper and Chen et al, 2022, and reached the following conclusion to my best effort. Although it is not as straightforward to argue that the regularization term lowers the discount factor (which makes it myopic and way less challenging), it does change the original objective into a surrogate one with a nonvanishing bias. I do not see how the induced bias can be mitigated because the regularization term can not be decayed into zero along the iterative path. In the current form, I tend to recommend rejection.

**Justification For Why Not Higher Score:**

See the above comments for how the regularization term changes the original objective, which is not thoroughly examined.

**Justification For Why Not Lower Score:**

N/A

---

### Decision · Program_Chairs · 2024-01-16

Reject